# Early Intensive Neurorehabilitation in Traumatic Peripheral Nerve Injury—State of the Art

**DOI:** 10.3390/ani14060884

**Published:** 2024-03-13

**Authors:** Débora Gouveia, Ana Cardoso, Carla Carvalho, Ana Catarina Oliveira, António Almeida, Óscar Gamboa, Bruna Lopes, André Coelho, Rui Alvites, Artur Severo Varejão, Ana Colette Maurício, António Ferreira, Ângela Martins

**Affiliations:** 1Arrábida Veterinary Hospital—Arrábida Animal Rehabilitation Center, 2925-538 Setúbal, Portugal; p60855@ipluso.pt (D.G.); anacardosocatarina@gmail.com (A.C.); mv.carla.c@gmail.com (C.C.); acatarinaroliv@gmail.com (A.C.O.); vetarrabida.lda@gmail.com (Â.M.); 2Superior School of Health, Protection and Animal Welfare, Polytechnic Institute of Lusophony, Campo Grande, 1950-396 Lisboa, Portugal; 3Faculty of Veterinary Medicine, Lusófona University, Campo Grande, 1749-024 Lisboa, Portugal; 4Faculty of Veterinary Medicine, University of Lisbon, 1300-477 Lisboa, Portugal; antonioalmeida@fmv.ulisboa.pt (A.A.); ogamboa@fmv.ulisboa.pt (Ó.G.); aferreira@fmv.ulisboa.pt (A.F.); 5Centro Interdisciplinar—Investigação em Saúde Animal (CIISA), Faculdade de Medicina Veterinária, Av. Universidade Técnica de Lisboa, 1300-477 Lisboa, Portugal; 6Centro de Estudos de Ciência Animal (CECA), Instituto de Ciências, Tecnologias e Agroambiente (ICETA), Universidade do Porto (UP), Rua D. Manuel II, Apartado 55142, 4051-401 Porto, Portugal; brunisabel95@gmail.com (B.L.); andrefmc17@gmail.com (A.C.); ruialvites@hotmail.com (R.A.); 7Departamento de Clínicas Veterinárias, Instituto de Ciências Biomédicas de Abel Salazar (ICBAS), Universidade do Porto (UP), Rua de Jorge Viterbo Ferreira, no. 228, 4050-313 Porto, Portugal; 8Associate Laboratory for Animal and Veterinary Science (AL4AnimalS), 1300-477 Lisboa, Portugal; avarejao@utad.pt; 9Cooperativa de Ensino Superior Politécnico e Universitário (CESPU), Avenida Central de Gandra 1317, 4585-116 Gandra, Portugal; 10CECAV, Centre for Animal Sciences and Veterinary Studies, University of Trás-os-Montes e Alto Douro (UTAD), Quinta de Prados, 5000-801 Vila Real, Portugal; 11Department of Veterinary Sciences, University of Trás-os-Montes e Alto Douro (UTAD), Quinta de Prados, 5000-801 Vila Real, Portugal

**Keywords:** PNI, neurorehabilitation, nerve repair, electrical stimulation, exercises, locomotor training, physical modalities

## Abstract

**Simple Summary:**

Peripheral nerve injuries are common in the clinical setting and may affect functionality by permanent monoplegia that could end in amputation. Electrical stimulation is an option to help recovery, in addition to exercises and locomotor training with a positive synergetic effect on nerve regeneration. These approaches could benefit from other rehabilitation techniques, such as laser and ultrasounds, as well as cell-based therapies, considering a One Health perspective.

**Abstract:**

Traumatic nerve injuries are common lesions that affect several hundred thousand humans, as well as dogs and cats. The assessment of nerve regeneration through animal models may provide information for translational research and future therapeutic options that can be applied mutually in veterinary and human medicine, from a One Health perspective. This review offers a hands-on vision of the non-invasive and conservative approaches to peripheral nerve injury, focusing on the role of neurorehabilitation in nerve repair and regeneration. The peripheral nerve injury may lead to hypersensitivity, allodynia and hyperalgesia, with the possibility of joint contractures, decreasing functionality and impairing the quality of life. The question remains regarding how to improve nerve repair with surgical possibilities, but also considering electrical stimulation modalities by modulating sensory feedback, upregulation of BDNF, GFNF, TrKB and adenosine monophosphate, maintaining muscle mass and modulating fatigue. This could be improved by the positive synergetic effect of exercises and physical activity with locomotor training, and other physical modalities (low-level laser therapy, ultrasounds, pulsed electromagnetic fields, electroacupuncture and others). In addition, the use of cell-based therapies is an innovative treatment tool in this field. These strategies may help avoid situations of permanent monoplegic limbs that could lead to amputation.

## 1. Introduction

Traumatic nerve injuries are common lesions that affect several hundred thousand humans every year in Europe and the United States of America [1,2,3]. The usual causes of the most severe cases include motor vehicle and sports accidents [2,4]. A similar reality is seen in veterinary medicine. Thus, in humans, as in dogs and cats, the brachial plexus injury (BPI) represents an example of peripheral nerve injury (PNI) that leads to devastating sensorimotor impairment of the affected limb in different possible degrees [2,5,6].

Clinical signs are the same in humans and in those species, and may include neuropathic pain, inability to bear weight and sensory deficits, which may be seen in, for example, nearly 57% of the cats that have radial and ulnar nerve injury [7]. Human and veterinary medicine can benefit mutually from research with this One Health perspective [8,9].

In this context, the assessment of nerve regeneration using animal models provides information for translational research and future therapeutic options that can be applied in human medicine [8,10].

Regarding Sunderland’s system, based on three different injuries, classification was according to the severity of nerve damage. In the first-degree injury (neuropraxia) there is a slight contusion with focal demyelination and interruption of impulse propagation, though the axons and nerve sheaths remain intact, with possible recovery in 3 months [11,12] and segmental myelinization in about 3 to 6 weeks [11,13]. The second injury (axonotmesis or crush injuries) leads to axon damage and removal of the Wallerian sheath with intact Schwann cells and endoneurium. In some cases, there is partially compressed endoneurium; however, perineurium and epineurium are still intact, allowing the regrowth of crushed axons [14]. Lastly, neurotmesis is a complete transection injury that manifests with damaged endoneurium and perineurium, although there is an intact epineurium with complete anatomical disruption of the nerve [12]. Thus, axonotmesis has a better prognosis than neurotmesis, though it is necessary time so that the muscle can regenerate from atrophy [15], so recovery can possibly extend to a period of 12–18 months [16,17].

Therefore, the purpose of this manuscript is to offer a hands-on vision of the non-invasive and conservative approaches for neurorehabilitation on PNI that could be implemented in a veterinary clinical setting. It also focuses on the role of neurorehabilitation in promoting strategies to promote nerve repair and regeneration, avoiding situations of permanent monoplegic affected limbs that could lead to amputation.

## 2. Diagnostic Assessment of Nerve Repair

The recovery of the nerve function after a lesion can be influenced by many factors, including the degree of nerve injury [14,15]. Functional recovery also depends on age, the time elapsed between injury and treatment, the distance from the injury site to the target muscle, the distance between the cell body and the site of axonal injury, as well as which part of the segmental nerve was affected [2,18] and on axonal reinnervation itself [19,20,21].

The degree of nerve injury itself depends on the number and size of fascicles damaged within the nerve [15]. Thus, depending on whether it is a partial lesion or a complete one, the nerve is not going to have the same ability for recovery (i.e., neuropraxia, axonotmesis and neurotmesis) [22,23,24,25,26]. So, there are several factors to be considered: type of injury, type of repair, timing of surgery, fascicle alignment and patient comorbidities [2,18]. The recovery and functional outcome can be also limited by inflammation, scar tissue and misdirection of regenerating sensory and motor axons, compromising the mechanisms of repair (i.e., remyelination, collateral sprouting, axon regrowth) [15,16,27]. All these are critical prognostic factors that may play an important role in regeneration [15,16,28,29].

Peripheral nerve regeneration is a complex process highlighted by Wallerian degeneration, axonal sprouting and remyelination [30]. The response to the injured nerve is influenced by damage degree and secondary degenerative changes, starting with a first degree based on a conduction blockage and mild degeneration. In the second degree, the calcium-mediated process known as Wallerian degeneration occurs distally to the injury site with few histological alterations [11]. The third-degree (intra-fascicular injuries) manifests by an evident local reaction, with elastic endoneurium retraction of the severed nerve fiber end. Consecutively, an inflammatory response begins with local hemorrhage and edema, developing a secondary fibrous scar that could lead to neuroma and perineural scar tissue [31].

Regeneration is estimated to a rate of nearly about 1 mm/day [30,32] and different nerve conduction studies (NCS) have been used [33,34,35,36,37,38], essentially, for the analysis of nerve conduction velocity [33,39] and electromyography (EMG), to record the compound muscle action potentials (CMAPs) [33,40,41,42], which is considered the best diagnosis to evaluate neuropathies and myopathies [43].

EMG and NCS are reported as the most sensitive tests for the evolution of axonal injury, allowing a more precise lesion location, which will enable monitoring and quantifying reinnervation or denervation [44,45]. Nerve conduction velocity measures impulse velocity along the motor neuron, and has a strong relationship with fiber morphology, during peripheral nerve regeneration [46]. Conduction velocity is proportional to fiber diameter; however, this association may be lost during neural regeneration since internodal lengths remain abnormally concise [44].

Therefore, regenerating and repairing an injured nerve occurs in multiple levels, which include the nerve cell body, the injury site, the segment between the neural cell body proximal stump and the segment between the injury site and the target organ [15,16].

Electrodiagnostic testing in the clinical setting of a physiotherapy and rehabilitation center is not realistically easy to apply. However, it is possible to evaluate the nociceptive withdrawal reflex as an outcome measure [33,47,48] and also to document weight [36,49,50,51,52] and muscle mass as a sign of an increase in fiber diameter [38,53,54,55,56,57]. Assessments of functional recovery may be possible by video-based motion analysis that makes a precise evaluation of muscle function during locomotion [41,58].

Gait analysis may assess normal and abnormal gait patterns, featuring nerve damage and measuring numeric comparisons [59]. Experimental research advancements in peripheral nerve regeneration, such as computerized video analysis, enable us to record footprints, walking patterns and bodyweight distribution, revealing the applicability of several parameters for locomotion evaluation [60]. Also, Schweizer et al. (2020) [61], in rodents, introduced an alternative approach to assess functional outcomes in sciatic damage and repair after the implementation of stem cells based on the swim test [62].

Furthermore, studies have shown monitoring of nerve regeneration by kinetic and kinematic evaluation of locomotion, as well as electrophysiologic and immunohistochemical outcomes [63]. Senger and colleagues (2020) [63] explored motor reinnervation assessment through kinetic and kinematic studies and skilled motor tasks, demonstrating that conditioning electrical stimulation (ES), with 20 Hz, lasting 1 h, over 7 days [64], increased Schwann cell proliferation in chronically injured nerves, enhancing axon regeneration and resulting in sensory and motor functional recovery.

Additionally, systemic biomarker measurements, such as the levels of neurotrophins and neurotrophic factors, could be used as nerve repair parameters [33]. The expression of brain-derived neurotrophic factor (BDNF) and its signal transduction receptor (TrkB) [33,47,65,66,67,68,69], as well as the nerve growth factor (NGF) and growth-associated protein 43 (GAP-43) levels, could be interesting parameters for this type of analysis [48,70]. After one week of intensive voluntary exercise, there should be an increase in the neurotrophic factor, BDNF [65,69,71]. In the clinical setting, functional recovery may be assessed by video-based motion analysis that precisely evaluates muscle function during locomotion [58,72].

On a daily basis, peripheral nerve injuries, for example, a BPI, are commonly diagnosed by history, clinical signs and neurological examination [73,74]. EMG can also help identify which nerves are affected. In regard to motor nerve conduction studies (MNCS), the association with EMG and often other tests (F waves, sensory nerve conduction studies, cord dorsum potentials), are commonly used in veterinary medicine to monitor patients during neurorehabilitation, which is already a common practice in human medicine [75,76].

However, in some complex cases, it is still insufficient to give the exact location of the injury. For example, in brachial plexus masses, magnetic resonance imaging could be helpful as a standard complementary exam [77]. In addition, these tests may require a specific timing that has to be considered [75,78]. In humans, they may not appear right after the injury; however, two–six weeks later, when fibrillations in denervated, muscles are present [22,79,80]. In dogs and cats, spontaneous pathologic activity at EMG appears 5 to 7 days after injury; nevertheless, all neuropathies are characterized by abnormal EMG activity [43].

## 3. The Problem of Neuropathic Pain

In human patients, traumatic PNI, such as BPI, is related to a higher incidence of neuropathic pain, when compared to other neuropathies (i.e., diabetic polyneuropathy, stroke, multiple sclerosis and refractory pain) [81,82,83,84].

Phantom limb pain is defined as pain that originates from the region of the former limb that is no longer physically part of the body [83,85]. This phenomenon is interpreted, in human medicine, as a reorganization of the cortical structures related to the areas that suffered the avulsion or amputation [81] and can be present in nearly 54–85% of amputees and 39.3% cases of nerve avulsion [81,86,87].

In dogs, phantom pain has been described to occur in about 60–80% of patients after amputation, within the first 2 years, and up to 10% may be present throughout their lifetime [88]. Thus, considering some animal models, to avoid self-mutilation a substance with discouraging flavor could be locally applied [59,89]. In clinical settings, it is advised to use an E collar to prevent automutilation, in addition to pharmacological treatment, usually with antiepileptic drugs, like gabapentin, pregabalin and sometimes tricyclic antidepressant drugs [90,91]. Normal doses range in gabapentin from 10 to 20 mg.kg TID [92,93,94], in pregabalin 4 mg.kg BID [95], in amitriptyline 3 to 4 mg.kg BID [96] and in amantadine 3 to 5 mg.kg BID or SID [97,98].

Neuropathic pain can develop after nerve injury due to atypical connections or incorrect interpretation of peripheral axons with the spinal cord through enhanced integration of sensory afferents with the spinal cord circuitry and preservation of the substance P levels [19,99]. To Woolf (1983) [100], there are central mechanisms, after PNI, that may lead to hypersensitivity, allodynia and hyperalgesia in response to mechanical/thermal stimuli [101,102]. The tingling, pulsating and pricking sensations are evoked by toxic activation of large Aβ and Aδ fibers associated with ectopic impulses in large fast conduction myelinated fibers [102,103]. The burning pain sensation may be due to intraneural stimulation of C nociceptive fibers [104]. Recent studies also suggest a role of the Schwann cells damage [105,106], in addition to the peripheral glia that could delay structural and functional changes following nerve injury [107].

Sodium channels are considered a main part of this phenomenon and an increased number of heterotopic channels (Na^+^ 1.8, Na^+^ 1.7 and Na^+^ 1.3) may decrease the stimulation threshold, resulting in neuropathic pain. Therefore, there are different underlying mechanisms, such as afferent nerves ectopic activity; peripheral sensitization; central sensitization; inaccurate inhibitory modulation; and pathologic activation of microglia [92,101].

This could be one of the explanations for why humans with BPI had reported an incidence of 30 to 80% of neuropathic pain [7,81,108,109] with high presentation of refractory pain [7,83,110,111,112]. In animal models, a study showed that in BPI, nearly 30% or more develop mechanical and cold allodynia [7,113], such as hyper-excitability of very superficial skin nociceptors [114], in which peripheral sensitization and maladaptive central changes occur, is usually associated with these allodynia and hyperalgesia [115]. Also, the high variability of sensory cutaneous innervation patterns is consistent with the pattern of weakness that can be observed. If the regeneration rate is near 1 mm per day, both distal nerve and muscles undergo atrophy and are unable to sustain regeneration for functional recovery [19] since the muscle force is further compromised by the incomplete recovery of the muscle fibers from denervation atrophy [116].

Immobilization promotes detrimental effects on the number of fibers of the muscles compared to the contralateral limb [33,53] and can delay recovery, most likely due to a reduction in muscle regeneration rate [33].

The nociceptive pain may be related to joint contractures, which are common in BPI, mostly in the carpus [117,118]. To avoid this type of injury, when approaching the sciatic nerve, studies have reported the application of kinesiotherapy passive exercises and physical activity stimulation in rat models. Examples of these were the implementation of a 45° inclined net to avoid long-term muscle inactivation [89,119]. In veterinary patients, there are several exercises to address joint contractures, such as assisted standing exercises, weight shifting, use of balance boards, walking through vertical weave poles, walking over cavalletti rails (Figure 1), and a variety of exercise rolls [120].

In human medicine, the modality of focal muscle vibration, which is a technique that applies vibratory stimuli to the muscles or their tendons using a mechanical device [121], may be an innovative non-invasive technique that intends to achieve neuroplasticity through enhancing changes in corticospinal excitability [121,122]. Among other treatments being studied, mostly for diabetic neuropathy, are transcranial direct current stimulation (tDCS) and repetitive transcranial magnetic stimulation (rTMS), in association with physiotherapy [121]. In addition, electroacupuncture, a traditional therapy for pain [123], and transcutaneous electrical nerve stimulation (TENS), a neuromodulation modality that has been widely used for symptomatic pain relief by the potential inhibition of activity and excitability of central nociceptive transmission neurons, can also be applied [124].

TENS is commonly used for dogs and cats in a clinical setting, intending to interfere with sensory perception and create an analgesic effect [125]. Also, in human patients, it is a common therapy for a wide variety of pain conditions [75,126].

In animal models, pulsed electromagnetic fields (PEMFs) have effects on relieving neuropathic pain in sciatic nerve chronic constriction injuries [127]. Pain perception may be influenced by PEMFs in animals and human behaviors. Scientific research explores whether there is a change in expression of HCN1/HCN2, after PEMF therapy. In peripheral nerves chronic constriction injuries, there is an increased spontaneous firing or changes in neurotransmitter conduction, resulting in chronic pain or persistent pain [128]. HCN2 expression in nociceptive neurons plays a key role in adjusting the generation of action potentials, in reaction to inflammation and the management of nociceptor excitability at a cellular level [129]. PEMFs should be more frequently used in clinical settings, though they have no effect on the change of expression of HCN1 and HCN2 mRNA after chronic constriction injuries. However, results indicate that nerve degeneration might be restored and improve locomotor functionality [130].

Electroacupuncture (EA) is a promising complementary strategy for neuropathic pain treatment, essentially through the descending inhibitory system and endogenous opioid peptides [131,132,133,134,135], such as β-endorphins, met-enkephalin and dynorphin [135]. This rehabilitation modality contributes to the upregulation expression of TRVT in the dorsal root ganglia (DRG) and spinal cord [131]. Xu and colleagues (2022) elucidated the regulation and role of the AMP-activated protein kinase/mammalian target of rapamycin (AMPK/mTOR) signaling pathway. AMPK/mTOR is involved in triggering autophagy in DRG macrophages, after EA stimulation in rats [131].

Zhang and colleagues (2014) associated neuropathic pain with BPI in rats, which is often not tolerated, as such injuries can be described as crushing, squeezing or burning [136]. EA function is to attenuate neuropathic pain after BPI, since this modality is considered safe, relatively unexpensive and easy to introduce in daily practice [136]. In dogs, it is crucial for the acupuncturist to undergo thorough training, which should focus on accurately locating each acupoint and developing a treatment plan according to traditional Chinese veterinary medicine (TCVM) work [134].

Hyperbaric oxygen therapy consists of the administration by inhalation of high doses of oxygen (100%), inside a hyperbaric chamber, that has a pressure that can range from 1.4 to 3 atmospheres absolute (ATA) [137]. This treatment favors the complete saturation of hemoglobin molecules in blood [138], increases oxygen dissolved in the plasma, promotes the production of the vascular endothelial growth factor (VEGF), decreases edema and promotes angiogenesis [139,140]. In perioperative peripheral nerve injuries in which tissue ischemia is the most common underlying mechanism of injury, it was suggested that hyperbaric oxygen therapy is a valuable procedure [141].

The authors use a standard protocol for the relief of neuropathic and nociceptive pain: the interferential TENS. This is achieved with two different channels and four rubber and carbon electrodes (7 × 5 cm) (BTL—4820 Smart^®^, Hertfordshire, UK), placed crossing each other at a 90° angle at the pain region or near the affected nerve pathway after the hair is clipped and gel applied. The programmed current is biphasic, symmetric and continuous, performed once or twice a day, 3–5 days a week, and sessions decreased according to pain evaluation. The current parameters are as follows: channel 1 (acute pain) with 80–150 Hz, until a maximum of 2.5 mA, pulse duration until 50 μs and time of treatment 10 min; channel 2 (chronic pain) with 10 Hz, until a maximum of 2 mA, pulse duration between 100 and 400 μs and time of treatment 10 min (Figure 2).

## 4. How to Improve PNI Repair?

Several non-surgical approaches, such as pharmacological, electrical, laser therapy and cell-based therapies, have been developed to promote remyelination and improve functional recovery in PNI [15,142,143,144,145]. For human patients, the most common treatment implies surgical resolution, and, in cases of a short gap (<1 cm), neurorrhaphy is frequently used with end-to-end sutures of the proximal and distal ends [8,25,146,147].

### 4.1. Surgical Approach

In regard to the surgical approach, for short gaps (<1 cm), the neurorrhaphy technique is used; however, it would cause excessive tension for a larger gap [8,25,146,147]. Thus, for medium and larger gaps, the most common technique is nerve grafting nerve reconstruction [8,148,149]. For gaps larger than 3 cm, the autograft is the current gold standard with an immunogenically inert scaffold that stimulates adhesion molecules and neurotrophic factors [8,25].

In human medicine, nerve transfers have been used as a reliable surgical option, preserving muscle and sensory innervation [150,151]. Also described were end-to-end nerve transfers for radial nerve palsies, traumatic ulnar nerve injury and ulnar nerve compressive neuropathy [152,153]. On the other hand, there is a case series describing vein wrapping after nerve repair [154].

Commercially nerve wraps based on collagen are available, such as NeuraWrap Integralife Sciences by bovine-derived type I collagen, already used for nerve repair of a rat sciatic nerve [155,156]. The AxoGuard nerve wrap (Axogen) by porcine small intestine submucosa was also already used [157,158]. The Hyaluronic acid–carboxy methylcellulose film (HA-CMC) and human amniotic membrane wrap are still in research [159,160,161].

Other alternative approaches are based on tissue engineering with the use of scaffolds and mesenchymal stem cells and their potential impacts, such as strength, biodegradability, biocompatibility, porosity, cell adhesion, differentiation, proliferation and growth [162].

#### Electrostimulation Modality

In the last years, electrostimulation (ES) has been shown to have the potential to enhance regeneration in different types of nerve injuries, including crush lesions [19,163,164], transection [165,166] and long-distance injuries [19,167]. This modality has been helping recovery in the context of One Health, due to its therapeutic mechanism to reduce muscle atrophy and promote active muscle reinnervation, increasing the expression of structural protective proteins and neurotrophic factors. Furthermore, it may possibly modulate sensory feedback and reduce neuralgia by inhibiting descending pathways [168].

Previous studies have suggested that the nerve effects of ES could be achieved by upregulating the expression of BDNF [66,169,170], glial cell-like derived neurotrophic factor (GDNF) [168,171], TrKB [172,173] and adenosine monophosphate (CAMP) [174].

The positive effects of ES in nerve repair were reported in both animals [33,34,41,51,67,166] and humans [33] (Figure 3). This efficient modality could maintain muscle weight, the twitch characteristics, modulating fatigue and mechanosensitivity [33,34,51].

Furthermore, functional electrical stimulation (FES) is an ES technique that uses sequences of short bursts of electrical pulses to stimulate nerves near the motor plate region or through peripheral afferent nerves, activating the peripheral spinal reflex. FES uses a low-intensity current, enough to trigger an action potential that induces muscle contraction. Low-frequency FES is used to promote nerve regeneration; however, the methods and frequencies applied diverge and need to be standardized due to the increase in nerve damage with high-frequency currents. Additionally, the use of biocompatible gels that provide skin maintenance and current uniform distribution on the electrodes makes this a better non-invasive stimulation approach with conventional surface electrodes suitable for innervating large muscles close to the skin [168,175].

FES has been shown to increase intraneuronal CAMP, improving regenerative ability via increased expression of the neurotrophins and cytoskeletal proteins [19]. Also, this modality may mimic a physiological wave of Ca^2+^ influx that generates a retrograde signal, leading to the activation of cell-autonomous mechanisms and promoting regeneration. BDNF, NGF and neurotrophins 4/5 may play an essential role in neuronal regeneration and maintenance [19,176,177,178]. These beneficial effects are associated with up-regulation of BDNF and its TrkB receptors in motoneurons [66,172].

Studies in both humans and animals have demonstrated that FES promotes preferential re-innervation of motor and sensory neurons, leading to a faster recovery [15,66,179], helping in the remyelination process [15,40] and avoiding nerve injury-induced muscle atrophy [15,180,181].

Most studies that are performed on animals resort to a low-frequency ES, usually 20 Hz [33,40,58,182,183] or 10 Hz [41,50], although a variable range of values from 20 to 200 Hz [184] or 4 to 75 Hz [34,51] has been documented.

The correct selection of frequency is mandatory because, as mentioned before, higher frequencies can deteriorate and aggravate atrophic muscle events [8,146,185]. Thus, to determine standard parameters, such as duration, it is important to consider the extent of damage variations of different injuries and possible side effects on the healthy tissues [8,146].

In human medicine, this modality can be also associated with surgical techniques [146,185]. For improving plantar spasticity, it was described a 5-day/week protocol [186] for 3–4 weeks, although longer treatments of 6–12 weeks could be necessary with pulse frequencies of 30–50 Hz and a pulse duration of 300 microseconds [186,187,188,189,190,191]. Treatment time has to consider potential fatigue, but usually ranges from 20 to 30 min per session [188,189,190].

Gunter and collaborators (2019) [192] stated that ES did not lead to neural damage when continuous stimulation with 20 Hz was applied for 16 h. Also, Agnew and McCreery (1990) [193] had several works demonstrating that ES was safe for application in the treatment of cats with PNI [194]. Furthermore, it was shown that a 20 Hz frequency was considered safe but, increasing to 50–100 Hz, even pulsed current and partial fiber recruitment could lead to neural damage.

Supporting this statement, Waters et al. (1985) [195] applied ES for 12 years in human patients with peroneal nerve lesions, using a frequency of 33 Hz, concluding that there was evidence of long-term safety with frequencies near 30 Hz.

The duty cycle describes the percentage of “on” and “off” stimulation time and it was shown that 50% of efficient stimulation time, with 50 Hz, could stimulate the peroneal nerve of a cat for a period of 16 h, with considerably less damage when compared to 100% stimulation time [193,196].

Thus, multiple animal studies defend the beneficial effects of ES with low frequencies and electrodiagnostic tests, revealing a high increase in CMAP scores after ES [197,198,199,200,201].

Actually, long-term stimulators have been surgically implanted in human patients, targeting nerves and securing electrode arrays [202]. These invasive devices penetrate the nerve to facilitate targeted activation of nerve fascicles [192,202].

Finally, ES may influence the concentration of circulating cytokines [203,204] and the modulation of neuroinflammatory response [205] through the macrophage and microglia action, which could be related to a temporary decrease in spasticity up to 40 min after treatments [206].

Additionally, ES could help in the four phases of PNI: oxidative stress stage (0–12 h); inflammation stage (12 h–3 days); atrophy stage (3–14 days) and atrophic fibrosis stage (14–28 days). The effects of treatment on peripheral neurogenesis vary according to the position of stimulation [207].

The author uses a standard protocol for the PNI that includes the following: until 50 Hz; duty cycle of 1:5; 10–16 mA; 10 min; and a trapezoid pulsated current (BTL—4820 Smart^®^, Hertfordshire, UK).

### 4.2. Exercises and Physical Activity

Even if there is not a clear relation between rehabilitation exercises and axon regeneration, there are specific physical exercises that avoid secondary lesions, such as disuse muscle atrophy, contracture, edema, stasis and pain. In human medicine, there is no developed standard treatment to be applied, as the used ones differ mostly in intensity duration and time [33].

However, exercises may promote angiogenesis, neurogenesis and neurotrophin expression, increasing nerve vascular integrity, decreasing apoptosis and modulating inflammation. Experimental findings, mostly in rodents, have shown the impact of exercise on synaptogenesis, myelination, neural recovery, growth development and muscle reinnervation. Examples are resorting to treadmill training [33,37,42,208], high-speed exercise running [56,209], swimming [36,210], voluntary locomotor exercises of endurance and resistance [71,211], isometric exercises [212], sensory retraining [213], manual stimulation [214], passive range of motion exercises and joint mobilizations [38,68].

The mechanisms of action related to the influence of physical exercises in nerve repair are different and based on research [33] resulting in evidence that this could be due to neurotrophin increments, such as BDNF and glial maturation factor (GMF), resulting in the survival and regeneration of damaged axons [48,67].

Thus, the effects of these exercises differ according to intensity and volume of training, as well as the type of nerve injury. For example, running on the land treadmill for 10 weeks could lead to faster nerve repair in rats [34], probably related to a reduced level of myelin-associated glycoprotein (MAG) on axonal growth inhibitor [10]. The MAG and complex gangliosides are related to long-term axon stability in both the central nervous system and peripheral nervous system [215], as a minor component of periaxonal myelin [216], allowing axoncytoarchitecture and regulating axon outgrowth [215], which is particularly important in human patients with peripheral nervous system immune diseases, such as Guillain–Barre Syndrome [217].

Axon regeneration development by treadmill training has been previously demonstrated and was shown in rats [10,35,218] and mice [10,37,219] with moderated daily training for 2 weeks [10].

The efficacy of exercise seems to start from the fourth regeneration week after nerve injury and not before that [33]. Therefore, overtraining and high workload could interfere with peripheral nerve recovery, mostly in the initial stages with detrimental effects [66], which could imply a physiological stimulus that interferes with anatomical and biochemical recovery [66,183]. However, sensory rehabilitation with intensive protocols could promote sensory perception [33,213].

Also, immobilization seems to have detrimental consequences on the count of the number of neural fibers, delaying repair because of a reduction in regeneration rate and not by the influence on nerve regeneration [33].

As expected, GDNF, BDNF and Insulin-like Growth Factor-1 (IGF-1) protein levels are increased in muscles that are exercised and may improve blood flow, activation of Schwann cells [220], leading to neovascularization, angiogenesis and enhanced metabolism rate [221]. In addition, it seems to have an impact on decreasing neuropathic pain and allodynia, but with poor positive effects described in humans [221].

Locomotor training could be one of the best options to help with PNI, starting with moderate exercise that increases in intensity and volume [8]. The BDNF, which is highly related to locomotor training, is also potentiated in association with electrical activity [66], promoting remyelination. The increase in neurotrophic factors by locomotor training may be limited when the distance between the axonal tops is estimated at 5 mm, as the neurotrophic factors could not improve regeneration, though they are generally increased by locomotor training [7].

Spontaneous peripheral nerve recovery is commonly inadequate and depends on the type of injury and damage extension [2]. A few studies in animals where moderate exercises and rehabilitation methods of motor and sensory functions were used reveal that such an approach could improve coordination and sensory–motor tasks [42], and the locomotor training could be an example of that in dogs, cats and humans [222] (Figure 4).

### 4.3. Combination of Electrical Stimulation and Locomotor Training

There is evidence in the current literature that, in human medicine, brief low-frequency electrical stimulation effectively promotes axon regeneration, maximizing functional recovery in PNI, namely in facial nerve stimulation with 20 Hz for 30 min/day after a crush injury, as well as after transection of the sciatic nerve with silicone tube and collagen gel surgical repair [223].

Thompson and collaborators (2014) [224] showed that moderate treadmill training and brief ES (with 20 Hz, for 1 h, in the sciatic nerve pathway) were applied in different groups of mice, for 5 days/week, revealing enhancement of axon repair. The same efficacy was proved in human patients submitted to carpal tunnel release due to medial nerve injury by constriction of the wrist ligaments [107]. Thus, these studies could be translated between human and veterinary medicine to improve recovery following injuries [179].

On the other hand, Elzinga et al. (2015) [225] showed that the same type of ES stimulated axon growth and muscle reinnervation after nerve surgery in rats and humans, improving regeneration in delayed nerve repair. In addition, activity-based exercises, such as a land treadmill combined with electrical stimulation after PNI, increase the potential of axon regeneration [66,165,224]. The same was reported with 20 Hz for 1 h in rats, mice [66,225] and human patients [226,227].

Prior investigations had addressed the role of ES in the complex pathophysiology of neuropathic pain, particularly in the inhibition of synaptic stripping and the excessive excitability of the dorsal roost ganglion, reducing pain and improving neurological function [207]. This modality could be safely used in conjunction with other treatments, such as pharmacological, cell-based therapies and rehabilitation techniques [19,207]. Everyday exercise with bipolar ES for 20 min significantly improved nerve regeneration and sensorimotor recovery, assessed by gait analysis, coordination tests and electrophysiological outcomes. Nowadays, there are human medicine clinical trials being conducted based on the effect of conditioning ES as a preoperative treatment prior to nerve decompression and reconstruction [228].

Early moderated and progressive training with electrical stimulation and locomotor training could help to reduce neuropathic pain [33,47], preventing neurotrophin-mediated hyperexcitability [33,209] and reducing facilitation of the monosynaptic H-reflex [33,37]. Additionally, this combination could be critical to enhance the chances of recovering mobility and avoiding secondary muscle or joint contractures [7].

According to Menchetti and colleagues (2020) [7], 25% of cats showed improved neurological condition with the support of physical therapies in a time scenario, which is considered fundamental. Thus, FES and treadmill exercise have been shown to have positive synergetic effects on nerve regeneration and muscle reinnervation [15].

The exact mechanism of ES and locomotor training to be implemented is still poorly established; however, CAMP and BDNF are reported to play a key role. ES could cause an increased influx of Ca^2+^ into the neurons followed by an increase in intracellular CAMP levels [15] and could be used as a rehabilitation intervention to stimulate and accelerate the process of nerve regeneration.

Implementation of these rehabilitation protocols in delayed time frames after PNI could, however, lead to different reactions on axon regeneration and motoneuron synapsis [35,99,197].

### 4.4. Other Rehabilitation Modalities

Low-level laser therapy (LLLT) is another modality that could be applied in a clinical setting after PNI [13]. This induces the upregulation of nitric monoxide, which is related to necrosis and apoptosis [229]. The nitric monoxide and other free radicals that result from lipidic peroxidation of the central and peripheral nervous systems may have an important role in neuropathic pain [230] and might be inhibited by laser therapy.

Low-power laser irradiation has clinical evidence in cats, regarding analgesic effects on peripheral nerves [231]. When applied to a dog’s spinal cord, glial scar formation decreases, with axonal sprouting by improving action potential on neuronal metabolism and synaptic transmission, essentially on the injured spinal cord region, which could improve restoration of ambulation [232].

This modality provides neuroprotection through overlapping mechanisms, including neuronal stimulation, neuromodulation and regeneration [233,234,235,236].

In terms of transcranial photobiomodulation (PBM), it was reported that the improvement of cerebral neurological function by ameliorating mitochondria dysfunction regulates the effects of apoptosis [237] and the antioxidant defense system [238].

LLLT could be combined with TENS, which was implemented in radial nerve injury of human patients, translating into significant effects compared with a control group, maintaining this improvement for 1–3 years [168]. Therefore, laser therapy was commonly applied in PNI every day for 5 consecutive days, followed by application once or twice a week, with positive effects [15].

These studies were based on the possible ability to promote regeneration and functional recovery of injured peripheral nerves, accelerating myelination, increasing axonal diameter, stimulating Schwann cell proliferation and improving motor nerve function [15,239,240,241]. The mechanisms behind this could be associated with DNA and RNA synthesis with consequent protein synthesis alongside cell proliferation, modifying nerve cell action potentials. The tissue biostimulation effects with possible increases in axonal diameter are important, although many issues arise due to the lack of standardized parameters [8].

The authors used a 980 nm laser, with a power of 10–19 mW/cm^2^ at the level of the PNI, for 3 to 5 days in a row, followed by 3 times per week, then 2 times, usually until 8 weeks of treatment (Figure 5).

The combination of laser therapy with ultrasounds is also a possibility, which has mechanical and thermal properties, stimulating blood circulation, release of BDNF and increase cell metabolism and tissue nutrition [8,185]. Some authors considered ultrasound to be more effective than LLLT in improving strength, pain and sensory deficits. However, according to Page et al. (2013) [242], there was no evidence of better results in using ultrasounds with the implementation of a splint, when compared to any other surgical procedure.

LLLT and EA also could be applied in association, since both approaches support tissue repair [233] and decrease inflammation, edema and fibrinogen levels [243,244,245]. LLLT and EA both promote additional analgesia by increasing endorphins synthesis [243,246]. The neuroprotection role is to support the remyelination of denuded axons [233,243,247]. These rehabilitation modalities (LLLT and EA) are safe, well-tolerated, relatively inexpensive and straightforward to learn and practice. In a clinical setting, it should be used early, as it provides a better opportunity to limit injury and maximize recovery [233].

For the authors, ultrasounds are usually performed with intensity ranging from 1 to 2.5 w/cm^2^, in a pulsed mode, with a duty cycle of 20% for 10 min. The recommended speed of the movement with the head sound over the skin should be slower but never static and mostly performed with longitudinal or circular patterns.

On the other hand, the use of LLLT plus splints has been shown to help in carpal contractures in human medicine [248].

Many studies demonstrated that PEMFs improve neuromuscular function in animals [249,250,251,252] and, in the long term, may determine synergistic functional progress [253]. PEMFs have been associated with larger fiber diameters, though with myelin thickness [249] and the decrease in intraneural fibrosis [252].

Scientific research has reported that PEMFs could have an ionic mechanism of action. There is also the possibility that PEMFs may increase cellular permeability to calcium ions, which has a key role in the upregulation of different biological processes [254]. This modality also increases growth factor local concentration, such as growth factor β_1_ (TGF-1), which increases TGF-1 level, and therefore should stimulate neuroglial signals and induce Schwan cell proliferation [254,255]. After PEMF treatment, we had an increase in BDNF and VEGF, which has an important role in axon distal regeneration [249].

Lee and colleagues (2023) showed positive outcomes regarding neural recovery when bone marrow mesenchymal stem cells were combined with PEMFs, suggesting that this association could be considered in future work on clinical applications [250].

EA is a therapeutic modality with a long-standing history of application in the Eastern world, where it has been used to treat a wide range of disorders over time.

Nowadays, EA is becoming commonly used in the Western hemisphere (United States), compared to Eastern countries (China and part of Europe) [134,256]. The clinical benefits of this modality in peripheral nerve injury are widely recognized [257,258], though the underlying mechanism remains incompletely understood.

EA improves nerve regeneration, which is possibly demonstrated by the effect of exosomal delivery of miR-21, which is strongly associated with nerve repair [259,260], and can mediate inflammation [260,261], oxidative stress [262], cell apoptosis [263], and proliferation [264]. EA used early would promote a better recovery in PNI [233].

Zeng and colleagues evidenced that EA can enhance nerve growth factor mRNA expression [265,266,267,268,269], insulin-like growth factor-1 [265,266,269] and neurotrophic factor 3, which could improve recovery from sciatic nerve crush injury in rats. EA has already been recognized as an effective modality to control pain and improve motor and sensory recovery in dogs with T3-L3 discopathies [134,270,271,272]. In dogs, EA association with surgery had better results in achieving ambulation [272]. Also, its combination with standard medical treatments in T3-L3 myelopathies, such as intervertebral disc disease in dogs, has been evaluated and culminated in an earlier return to ambulation, when compared with standard medical treatment alone, which can be justified possibly due to S 100β level increase. The release of additional neurotrophic factors that promote regeneration may occur following EA [273].

The use of EA in association with stem cells in rats with spinal cord injury (SCI) has improved the regeneration, survival and migration of transplanted stem cells towards injury [256,274]; nevertheless, more studies should be performed.

EA could increase BDNF expression, which will reduce neuronal death, provide a suitable microenvironment for nerve development [275] and upregulate glial cell-derived neurotrophic factor (GDNF), essentially in facial motoneurons [276]. Low-frequency EA could improve sciatic nerve regeneration [40,277,278,279,280,281].

### 4.5. Cell-Based Therapies and PNI

Cell-based therapies are the most innovative treatment approaches in the PNI field. This can help damaged tissues by targeting differentiation processes that influence cell morphology, metabolic activity, growth factors secretion and signal response [2,30].

Stem cells can help in nerve regeneration by promoting a neuroprotective microenvironment that modulates degeneration and apoptosis, supporting neurogenesis, axonal growth and remyelination [282] (Figure 6). Increased cell metabolism could be also related to an increase in neurotrophin 3 (NT-3), neurotrophin 1 (NT-1), neurotrophin 4 (NT-4), ciliary-derived neurotrophic factor (CDNF), BDNF, NGF and GDNF 4 [282,283,284].

In addition, stem cells, such as mesenchymal stem cells (MSCs), may increase neovascularization and promote secretion of tissue inhibitor of metalloproteinase—8 (VEGF), angiopoietin 1 and transformation of growth factor B and IL-8 [282,284].

The MSCs have a paracrine role in modulating neuroinflammation and immune response. This immunomodulated response occurs through pro-inflammatory cytokines produced by lymphocytes and can activate MSCs. Thus, MSC can inhibit scar tissue formation, promoting angiogenesis and tissue regeneration [2,8].

Furthermore, MSCs have self-renewal properties and are able to differentiate into neural-like and Schwann-like myelinating cells [2,8]. They display a role in decreasing the expression of pro-apoptotic factors while potentiating anti-apoptotic mechanisms [285].

## 5. Conclusions

Recovery and repair of nerve sensory–motor functions depend on several different factors. However, when regeneration happens, it still depends on a rate of around 1 mm/day, leading to muscle atrophy, joint contractures, persistent lameness, a weakness that enables weight support and possible neuropathic pain. Thus, non-invasive neurorehabilitation modalities could be prescribed in PNI, considering the synergetic power of FES and locomotor training as one of the best therapeutic approaches to obtain faster recovery of sensory–motor functions. It also plays an essential part in avoiding neurogenic atrophy and secondary muscle or joint contractures, which will support reinnervation.

Essentially, these protocols associated with regenerative medicine, including stem cell transplantation (such as MSCs), are innovative therapeutic tools in the field of nerve repair that may help to revert cellular changes, reducing neural apoptosis and supporting neurogenesis (Figure 7). Further studies with similar protocols for PNI should be conducted based on a One Health perspective.

## Figures and Tables

**Figure 1 animals-14-00884-f001:**
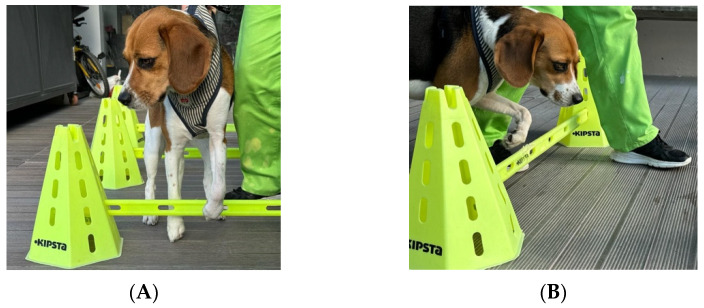
Walking over cavalletti rails (**A**,**B**), on a patient with BPI, neuropraxia and axonotmesis of the radial nerve.

**Figure 2 animals-14-00884-f002:**
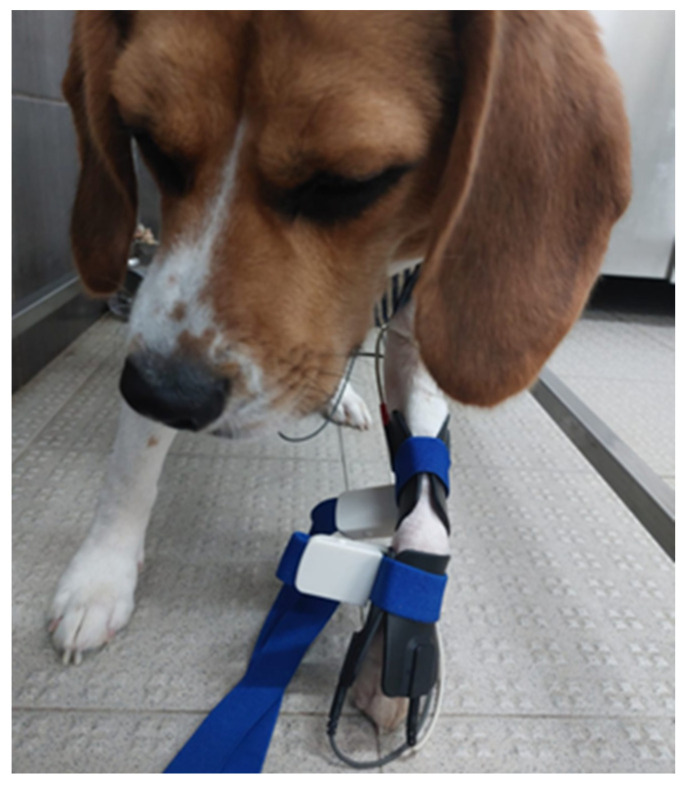
Interferential transcutaneous electrical nerve stimulation applied in the carpus of a dog with neuropathic pain.

**Figure 3 animals-14-00884-f003:**
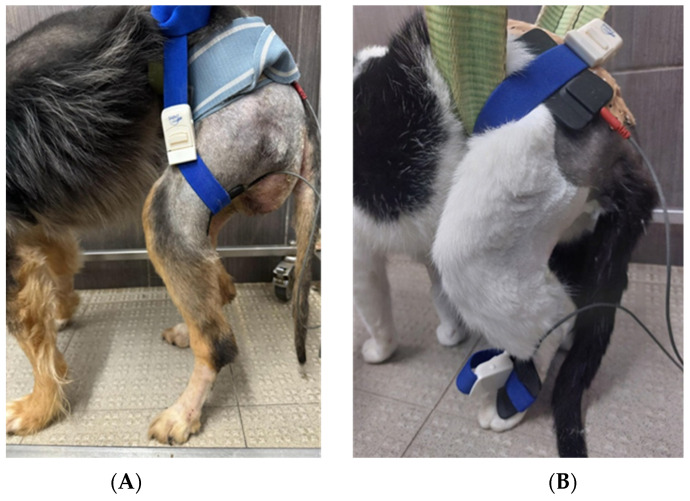
Functional electrical stimulation on the flexor muscle group of the hindlimb applied on a dog (**A**) and a cat (**B**).

**Figure 4 animals-14-00884-f004:**
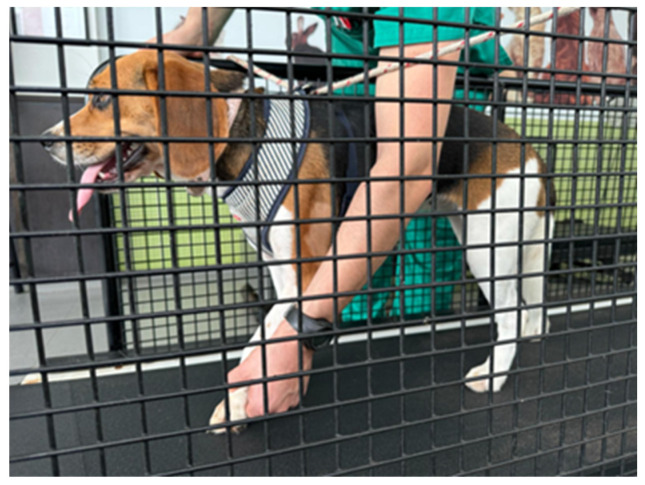
Land treadmill locomotor training with bicycle movements performed in a dog with monoplegia.

**Figure 5 animals-14-00884-f005:**
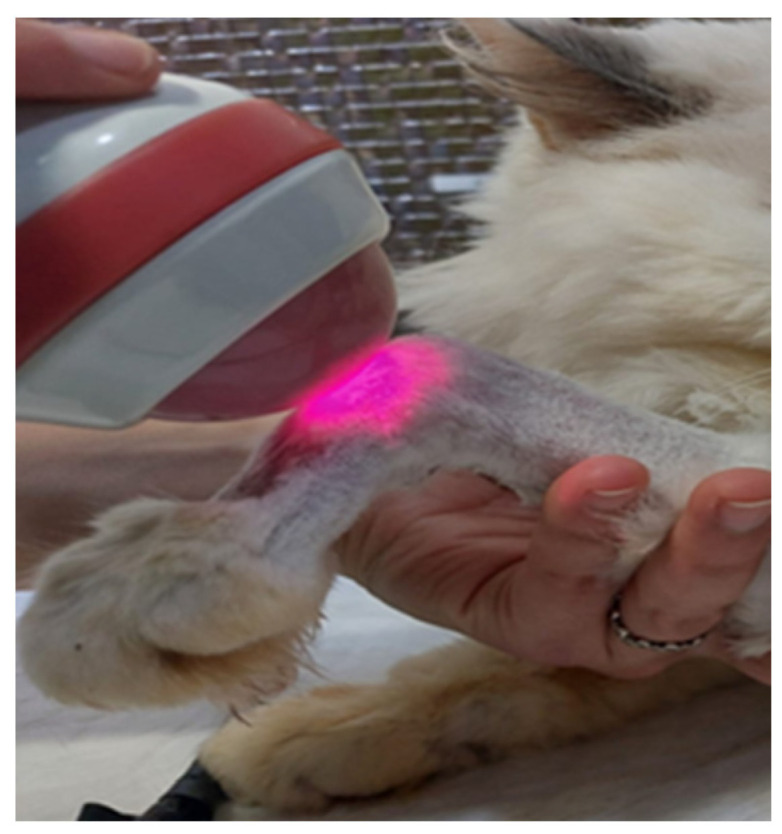
Laser therapy applied to the carpus of a cat with a brachial plexus injury.

**Figure 6 animals-14-00884-f006:**
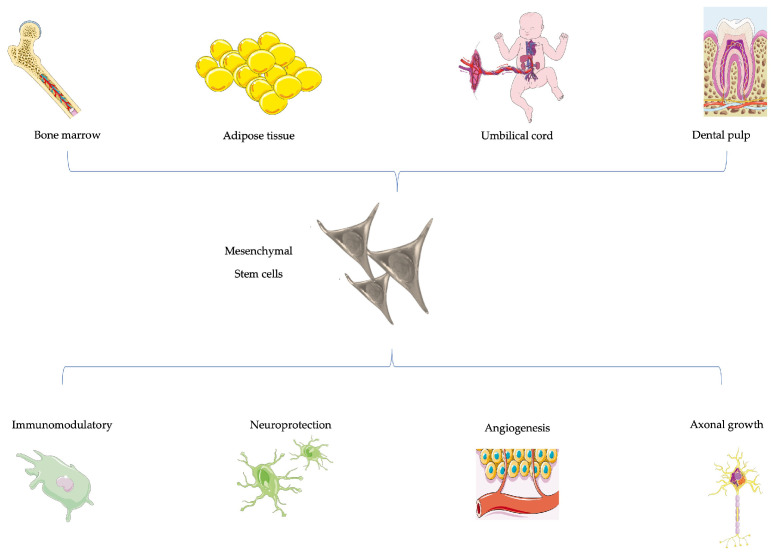
Mesenchymal stem cell-based therapies and their beneficial effects.

**Figure 7 animals-14-00884-f007:**
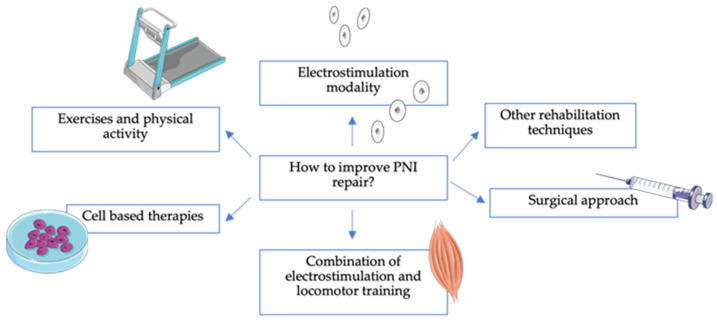
Multimodal protocol approach to improve peripheral nerve injury repair.

## Data Availability

The data presented in this study are available upon request from the corresponding author.

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
