# Peer review of "Early Intensive Neurorehabilitation in Traumatic Peripheral Nerve Injury—State of the Art"

_animals, 2024, doi:10.3390/ani14060884_

Round 1

Reviewer 1 Report

Comments and Suggestions for Authors

The paper is a review of physiotherapeutic approaches to dogs and cats with traumatic peripheral nerve injury. A complete and detailed description of the most important treatment modalities is given, based on a very complete bibliography and on the direct experience of the authors.

Areas of weakness in this paper are those relating to clinical assessment, diagnosis and other treatment modalities. The text, especially in these parts of the article, is not always clear and easy to follow. Despite the completeness of references (especially in human medicine), information and data on some diagnostic procedures in animals are incomplete or incorrect. I suggest integrating (and possibly replacing some) human references with publications on nerve injury in domestic animals.

At least a minimum of information regarding the use in small animals with suspect nerve injury of electrodiagnostic studies and drugs for neuropathic pain should be added.

Specific comments:

Line 39. upregulation BDNF, GFNF, TrKB. upregulation of BDNF...?

Lines 62-64. The sentence is unclear, please rewrite it

Lines 70-72. This sentence is incorrect. There is probably a verb missing.

Lines 83-83 and 90-91. The same concept is repeated. Please try to remove this repetition.

Lines 108-110. The sentence in the present form is unclear. Please try to explain how the evaluation of conduction velocity and CMAP during motor nerve conduction studies (MNCS) can provide information regarding regeneration rate. Compound muscle ACTION potentials (CMAPs)

Lines 145-149. I would add that electromyography (EMG) is another electrodiagnostic test that can help identify which nerves are affected. With regard to MNCS, this test, together with EMG and often other tests (F waves, sensory nerve conduction studies, cord dorsum) is commonly used in veterinary medicine (dogs, cats etc.) to monitor the patients during neurorehabilitation.

Lines 152-154. This is not true in dogs and cats. Spontaneous pathologic activity at EMG appears 5-7 days after injury. Please replace these 5 citations with one relevant reference on veterinary neurology.

Lines 166-167. I suggest to add the use of E collar to prevent automutilation. I am not aware of any evidence of the efficacy of substances with a deterrent flavour in dogs or cats with neuropathic pain. A sentence emphasising the importance of medications for neuropathic pain (gabapentin, pregabalin, etc.) should be included in this paragraph.

Lines 186-187. I do not understand from which of these 3 publications these data have been taken.

Lines 189-190. Please clarify this sentence. Do you mean that regeneration inevitably does not lead to functional recovery?

Lines 194-196. The proposed pathogenesis of hyperflexion of distal joints in denervation is not widely accepted. I suggest to remove the second part of this sentence.

Line 214. Correct punctuation

Line 254. Insert …, electrostimulation (ES)…

Lines 379-380. The sentence is unclear, please rewrite it

Line 401. On the other hand

Author Response

REVIEWER #1

Comments and Suggestions for Authors

The paper is a review of physiotherapeutic approaches to dogs and cats with traumatic peripheral nerve injury. A complete and detailed description of the most important treatment modalities is given, based on a very complete bibliography and on the direct experience of the authors.

Areas of weakness in this paper are those relating to clinical assessment, diagnosis and other treatment modalities. The text, especially in these parts of the article, is not always clear and easy to follow. Despite the completeness of references (especially in human medicine), information and data on some diagnostic procedures in animals are incomplete or incorrect. I suggest integrating (and possibly replacing some) human references with publications on nerve injury in domestic animals.

We kindly appreciate the reviewer´s comments and taking all in consideration. We made all the changes suggested.

At least a minimum of information regarding the use in small animals with suspect nerve injury of electrodiagnostic studies and drugs for neuropathic pain should be added.

We thank the reviewer for this comment and we introduced the information suggested.

Specific comments:

Line 39. upregulation BDNF, GFNF, TrKB. upregulation of BDNF...?

We kindly appreciate the reviewer’s comments, and as suggest we introduced “of BDNF”.

Lines 62-64. The sentence is unclear, please rewrite it

We kindly appreciate the reviewer’s comments, and we corrected the sentence.

Lines 70-72. This sentence is incorrect. There is probably a verb missing.

We thank the reviewer for this comment, and introduced the verb that was missing.

Lines 83-83 and 90-91. The same concept is repeated. Please try to remove this repetition.

We thank the reviewer for this comment, and we adapted the sentence and removed the repetition, so that the concept was kept.

Lines 108-110. The sentence in the present form is unclear. Please try to explain how the evaluation of conduction velocity and CMAP during motor nerve conduction studies (MNCS) can provide information regarding regeneration rate. Compound muscle ACTION potentials (CMAPs)

We appreciate your comment, we remodeled the sentence, corrected “Compound muscle action potentials (CMAPs)” and created a clarifying paragraph.

Lines 145-149. I would add that electromyography (EMG) is another electrodiagnostic test that can help identify which nerves are affected. With regard to MNCS, this test, together with EMG and often other tests (F waves, sensory nerve conduction studies, cord dorsum) is commonly used in veterinary medicine (dogs, cats etc.) to monitor the patients during neurorehabilitation.

As suggested by the reviewer, we have rewrite  the sentence with the recommendations.

Lines 152-154. This is not true in dogs and cats. Spontaneous pathologic activity at EMG appears 5-7 days after injury. Please replace these 5 citations with one relevant reference on veterinary neurology.

Thank you so much for your comment, we did the correction and adapted the sentence.

Lines 166-167. I suggest to add the use of E collar to prevent automutilation. I am not aware of any evidence of the efficacy of substances with a deterrent flavor in dogs or cats with neuropathic pain. A sentence emphasizing the importance of medications for neuropathic pain (gabapentin, pregabalin, etc.) should be included in this paragraph.

We appreciate your comment, we modified the paragraph and we added your suggestions.

Lines 186-187. I do not understand from which of these 3 publications these data have been taken.

We thank the reviewer for this comment, in this sentence: “In animal models, a study had showed that in BPI, …”  we referred to rats and this information was taken from reference 7, and was supported by Wang, L. When we referred the reference 113, it was to prove that cold allodynia exists, however we are going to remove it, since it refers to a study on humans in a hospital setting.

Lines 189-190. Please clarify this sentence. Do you mean that regeneration inevitably does not lead to functional recovery?

We kindly appreciate the reviewer´s comment, considering a normal muscle mass, what we see in clinical practice is that it is inevitable, because these patients are waiting for spontaneous recovery. What was intended to be said is that because regeneration is so slow, the process over time will cause severe muscle atrophy, that it is considered that recovery may not be functional. It is thought that fewer regenerated axons allow less than 50% of the muscle fibers in these muscles, and this is associated with a progressive decline in muscle force.

Lines 194-196. The proposed pathogenesis of hyperflexion of distal joints in denervation is not widely accepted. I suggest to remove the second part of this sentence.

As suggested by the reviewer we remove the second part of the sentence.

Line 214. Correct punctuation

We kindly appreciate the reviewer’s comment, therefore with erased the dot (.) in the middle of the sentence.

Line 254. Insert …, electrostimulation (ES)…

We thank the reviewer for this comment, and we added “eletrostimulation (ES)”.

Lines 379-380. The sentence is unclear, please rewrite it

Thank you so much for your comment we rewrite the sentence.

Line 401. On the other hand

We appreciate the reviewer comment, and we corrected the beginning of the sentence.

Reviewer 2 Report

Comments and Suggestions for Authors

The manuscript has potential but needs extensive revision.

Lines 49-54 please quote original source scientific articles, not a quote of a reference from these papers.

Lines 72-74 please reword, difficult to make sense of with current sentence structure

Line 82 – Add reference 10 also

Lines 86-87 needs a reference

Lines 101-102, needs rewording, difficult to make sense of with current sentence structure

Line 109 – the word “were” should be replaced with ‘have been’

Lines 116-118 “muscle mass by increase in fiber diameter” should be replaced with “muscle mass as a sign of increase in fiber diameter”

Line 127 specify in rodents

Line 132 – how often and what were the settings of ES?

Lines 141-143 do not flow as a sentence

Line 144 “is” should be replaced with “are”

Lines 166-167, sentence does not make sense, needs rewording, also important to mention pain management as a way to avoid self mutilation (particularly pregabalin, amantadine etc)

Line 168 sentence should start with “neuropathic” not ‘the”

Lines 187-189 – please expand

Line 194 “and is” should be replaced with “which is”

Lines 198-199 – please explain relevance of this exercise to veterinary patients vs. lab animals

Lines 204-205 – transcranial – do the authors mean transcutaneous?

Line 211 (reference 109) TENS may be important in the clinic the article is from but is largely replaced by PEMF and electroacupuncture in US veterinary rehabilitation. Avoid referring to commonality if there are no data numbers to back this up beyond a single clinic.

The authors have omitted PEMF laboratory studies and the studies done in a clinical setting on dogs recovering from back surgery (research, not review papers) Seem to be focused on lab animal research and human medicine vs clinical research. Electroacupuncture is omitted also as well as hyperbaric oxygen therapy 

Line 214 “to the relief neuropathic” should read “for the relief of neuropathic”

Lines 336-337 specify in human medicine

Lines 390-394 state what species

Lines 452-454 – the authors make a recommendation but no research basis or references are made for this recommendation 980nm has variable penetration into tissues.

Other references (authors should read these and I recommend searching further)

https://pubmed.ncbi.nlm.nih.gov/29099681/

https://pubmed.ncbi.nlm.nih.gov/33195591/

https://pubmed.ncbi.nlm.nih.gov/30879228/

https://pubmed.ncbi.nlm.nih.gov/1351254/

https://pubmed.ncbi.nlm.nih.gov/31571661/

https://pubmed.ncbi.nlm.nih.gov/15910181/

https://pubmed.ncbi.nlm.nih.gov/10146265/

https://pubmed.ncbi.nlm.nih.gov/33477408/

https://pubmed.ncbi.nlm.nih.gov/37413829/

https://pubmed.ncbi.nlm.nih.gov/21061457/

https://pubmed.ncbi.nlm.nih.gov/30703615/

https://pubmed.ncbi.nlm.nih.gov/17867976/

https://pubmed.ncbi.nlm.nih.gov/12819372/

https://pubmed.ncbi.nlm.nih.gov/20513202/

https://pubmed.ncbi.nlm.nih.gov/26464906/

https://pubmed.ncbi.nlm.nih.gov/19405884/

https://pubmed.ncbi.nlm.nih.gov/33132818/

https://pubmed.ncbi.nlm.nih.gov/31477463/

https://pubmed.ncbi.nlm.nih.gov/36605615/

https://pubmed.ncbi.nlm.nih.gov/36660615/

https://pubmed.ncbi.nlm.nih.gov/29654733/

https://pubmed.ncbi.nlm.nih.gov/33334124/

https://pubmed.ncbi.nlm.nih.gov/37672149/

https://pubmed.ncbi.nlm.nih.gov/31996011/

https://pubmed.ncbi.nlm.nih.gov/35132352/

https://pubmed.ncbi.nlm.nih.gov/25221593/

https://pubmed.ncbi.nlm.nih.gov/33797305/

https://pubmed.ncbi.nlm.nih.gov/37277911/

https://pubmed.ncbi.nlm.nih.gov/27901476/

https://pubmed.ncbi.nlm.nih.gov/15605398/

https://pubmed.ncbi.nlm.nih.gov/25448645/

https://pubmed.ncbi.nlm.nih.gov/37124111/

https://pubmed.ncbi.nlm.nih.gov/1984256/

https://pubmed.ncbi.nlm.nih.gov/6603461/

https://pubmed.ncbi.nlm.nih.gov/6695016/

Comments on the Quality of English Language

See above

Author Response

REVIEWER #2

Comments and Suggestions for Authors

The manuscript has potential but needs extensive revision. 

Lines 49-54 please quote original source scientific articles, not a quote of a reference from these papers.

Thank you so much for your comment we quote the original source scientific articles.

Lines 72-74 please reword, difficult to make sense of with current sentence structure

As suggested by the reviewer, we changed the sentence structure.

Line 82 – Add reference 10 also

We appreciate your comment, we add the reference.

Lines 86-87 needs a reference.

We appreciate your comment and we introduced the reference.

Lines 101-102, needs rewording, difficult to make sense of with current sentence structure

We are very grateful for the reviewer´s comment, and we changed the sentence structure.

Line 109 – the word “were” should be replaced with ‘have been’

We thank the reviewer for this comment, and we corrected the word “were”.

Lines 116-118 “muscle mass by increase in fiber diameter” should be replaced with “muscle mass as a sign of increase in fiber diameter”

As suggested by the reviewer we modified the sentence.

Line 127 specify in rodents

We appreciate your comment and introduced your recommendation in the test.

Line 132 – how often and what were the settings of ES?

Thank you so much for your comment we add the frequency and settings of ES.

Lines 141-143 do not flow as a sentence

As suggested by the reviewer we restructure the sentence.

Line 144 “is” should be replaced with “are”

We are very grateful for the reviewer comment, and we replaced the verb.

Lines 166-167, sentence does not make sense, needs rewording, also important to mention pain management as a way to avoid self-mutilation (particularly pregabalin, amantadine etc)

Thank you so much for your comment, as you suggested we reword and restructured the paragraph.

Line 168 sentence should start with “neuropathic” not ‘the”

We appreciate your comment, we remove “the”.

Lines 187-189 – please expand

We thank the reviewer so much for this comment, and we followed your suggestion.

Line 194 “and is” should be replaced with “which is”

As suggested by the reviewer we corrected the sentence.

Lines 198-199 – please explain relevance of this exercise to veterinary patients vs. lab animals

We are very grateful for this comment, and we introduced the exercises that are performed in veterinary medicine.

Lines 204-205 – transcranial – do the authors mean transcutaneous?

We are very grateful for the reviewer’s comment, and we mean transcranial as it is in the text.

Line 211 (reference 109) TENS may be important in the clinic the article is from but is largely replaced by PEMF and electroacupuncture in US veterinary rehabilitation. Avoid referring to commonality if there are no data numbers to back this up beyond a single clinic.

The authors have omitted PEMF laboratory studies and the studies done in a clinical setting on dogs recovering from back surgery (research, not review papers) Seem to be focused on lab animal research and human medicine vs clinical research. Electroacupuncture is omitted also as well as hyperbaric oxygen therapy  

Thank you so much for your comment, as you suggested we introduced more information regarding PEMF, electroacupunctures and hyperbaric oxygen therapy.

In perioperative peripheral nerve injuries, in which the tissues ischemia is most common underlying mechanism of injury, it was suggested that hyperbaric oxygen therapy is valuable procedure.

Line 214 “to the relief neuropathic” should read “for the relief of neuropathic”

As suggested by the reviewer we corrected the word.

Lines 336-337 specify in human medicine

We thank the reviewer so much and followed your suggestion.

Lines 390-394 state what species

As suggested by the reviewer we indicate in which specie.

Lines 452-454 – the authors make a recommendation but no research basis or references are made for this recommendation 980nm has variable penetration into tissues.

We thank the reviewer for your comment, when evaluating the application of 980 nm wavelength for neurologic injury, the literature is equivocal. However according to Piao and colleagues (2019), this wavelength could be indicated. Nevertheless, the neuropathic changes that are induced by the photobiomodulation at 980 nm transcutaneous, include regeneration of the intra-epidermal nerve fibers, re-innervation of the Langerhan cells and decrease in expression of protein gen product 9.5 (22). In vitro exposure of neuronal cells to 980 nm, laser has been shown to modulate sodium channel proteins by laser induced photothermal effect (23) and improve neurite elongation (primary rat cortical neurons), by non-thermal means (24). We perform this around 14 years, with a Companion LASER, and never had any trouble with it, and also with this improves neuropathic pain.

Round 2

Reviewer 2 Report

Comments and Suggestions for Authors

The manuscript is much improved, a big thank you to the authors for tackling this subject and being open to revision. 

The self-citation still inflates the importance of using 980nm laser and the laser power is very high (line 970). Would prefer "in the authors practice....... is used". Remove the word 'essential'.

Comments on the Quality of English Language

The cadence and wording of the manuscript is very good, just a few minor issues that the editor will catch (mouses = mice for example). 

Author Response

REVIEWER #2

Comments and Suggestions for Authors

The manuscript is much improved, a big thank you to the authors for tackling this subject and being open to revision.

The self-citation still inflates the importance of using 980nm laser and the laser power is very high (line 970). Would prefer "in the authors practice....... is used". Remove the word 'essential'.

Answer: Dear Reviewer, we kindly appreciate all your efforts in this revision that contributed to a great improve in our manuscript. As suggested, we changed the sentence regarding the laser and corrected the word “mouses”. Thank you once again for your contribution.
